**Investigation**

# Hi-reComb: constructing recombination maps from bulk gamete Hi-C sequencing

Milan Malinsky [iD],[1,2,*] Marion Talbi [iD],[1,2] Chenxi Zhou [iD],[3] Nicholas Maurer,[4,5] Samuel Sacco,[4,5] Beth Shapiro,[4,5] Catherine L. Peichel [iD],[1] Ole Seehausen,[1,2] Walter Salzburger,[6] Jesse N. Weber,[7] Daniel I. Bolnick [iD],[8] Richard E. Green [iD],[4,5] Richard Durbin [iD] [8]

[1]Institute of Ecology and Evolution, University of Bern, Bern 3012, Switzerland
[2]Department of Fish Ecology and Evolution, EAWAG, Kastanienbaum 6047, Switzerland
[3]Department of Genetics, University of Cambridge, Downing Street, Cambridge CB2 3EH, United Kingdom
[4]Department of Biomolecular Engineering, University of California Santa Cruz, Santa Cruz, CA 95064, United States
[5]UCSC Genomics Institute, University of California Santa Cruz, Santa Cruz, CA 95064, United States
[6]Department of Environmental Sciences, Zoological Institute, University of Basel, Basel 4051, Switzerland
[7]Department of Integrative Biology, University of Wisconsin-Madison, Madison, WI 53706, United States
[8]Department of Ecology and Evolutionary Biology, University of Connecticut, Storrs, CT 06269, United States

*Corresponding author: Institute of Ecology and Evolution, University of Bern, Baltzerstrasse 6, Bern 3012, Switzerland. Email: millanek@gmail.com

Recombination is central to genetics and to evolution of sexually reproducing organisms. However, obtaining accurate estimates of recombination rates, and of how they vary along chromosomes, continues to be challenging. To advance our ability to estimate recombination rates, we present `Hi-reComb`, a new method and software for estimation of recombination maps from bulk gamete chromosome conformation capture sequencing (Hi-C). Simulations show that `Hi-reComb` produces robust, accurate recombination landscapes. With empirical data from sperm of five fish species we show the advantages of this approach, including joint assessment of recombination maps and large structural variants, map comparisons using bootstrap, and workflows with trio phasing vs. Hi-C phasing. With off-the-shelf library construction and a straightforward rapid workflow, our approach will facilitate routine recombination landscape estimation for a broad range of studies and model organisms in genetics and evolutionary biology. `Hi-reComb` is open-source and freely available at https://github.com/millanek/Hi-reComb.

Keywords: recombination map; genetics; Hi-C; gametes; sperm; software; cichlids; stickleback

## Introduction

Meiotic recombination is a hallmark of sexual reproduction. While mutations give rise to new genetic variants, recombination shuffles them to generate new haplotypes—that is, chromosomes with novel combinations of existing alleles which selection can act on. Therefore, recombination impacts key evolutionary processes such as adaptation and speciation, and it shapes the distribution of genetic variation along the genomes. Because organismal traits are usually influenced by many genetic variants (Shi et al. 2016; Boyle et al. 2017; Barton 2022), often with nonlinear epistatic interactions among them (Phillips 2008; Domingo et al. 2019; Johnson et al. 2023), and because adaptation and speciation commonly require co-evolution of a whole suite of traits (Phillips and Arnold 1989; White and Butlin 2021), evolution is increasingly seen as multidimensional and combinatorial (Marques et al. 2019; Barton 2022), with recombination in a central role. Yet, routine and accurate genome-wide reconstruction of recombination maps remains a major challenge, and there is a need for novel approaches and methods for recombination inference.

Recombination rates vary by several orders of magnitude along the genome (Coop and Przeworski 2007; Stapley et al. 2017; Halldorsson et al. 2019). Accurate inference and representation

of this variation requires ascertaining the chromosomal locations of many crossover events. Depending on how this is done, recombination rate inference methods can be divided into three categories (Peñalba and Wolf 2020): (i) indirect population genetic approaches based on patterns of linkage disequilibrium (LD); (ii) genetic linkage maps based on the transmission of polymorphic markers in crosses and pedigrees; and (iii) gamete sequencing, based on finding breakpoints in gamete haplotypes compared to the donor genome. The recombination maps inferred by the three approaches differ in several important respects, perhaps most notably in resolution, the measured time interval, and in how they are impacted by selection (Peñalba and Wolf 2020). For example, LD-based maps typically integrate over thousands of generations of recombination, deliver high resolution, but are influenced by selection and other population genetic processes affecting LD (e.g. population size changes) occuring over that timeframe. In contrast, studies using crosses and pedigrees directly reflect crossovers passed across one or a small number of generations but rarely include the thousands of samples needed to achieve resolution comparable to LD-based maps [but see (Morgan et al. 2017; Halldorsson et al. 2019) for high resolution maps in humans and mice]. Finally, approaches based on

sequencing of sperm, eggs, or pollen reveal all crossovers that occured during successful gametogenesis, and thus deliver a direct snapshot of the recombination process, not affected by selection acting at other life stages. The resolution of these gamete-based maps is largely determined by the number of gametes that can be typed.

Single sperm whole-genome sequencing has been used for well over a decade (Lu et al. 2012; Wang et al. 2012) and continues to deliver insights into the factors that influence meiotic recombination (Hinch et al. 2019; Yang et al. 2022). However, practical limitations have restricted studies to typing at most a few hundred sperm cells. One exception is the Sperm-seq (Bell et al. 2020) protocol—a sperm specific variation of high-thoughput single cell sequencing (Macosko et al. 2015)—which scaled to over 30,000 sperm cells. Nevertheless, the adoption of this approach beyond the original study appears to be limited, likely due to the custom and relatively complex laboratory procedure that is required. Bulk sequencing approaches avoid the need for the isolation of individual cells. Three studies introduced linked-read sequencing of bulk sperm and pollen (Dréau et al. 2019; Sun et al. 2019; Xu et al. 2019), demonstrating that single cell sequencing of individual gametes is not necessary to infer crossover events. These studies use read-linkage information to find recombination breakpoints by comparing individually barcoded gamete DNA fragments against the haplotypes of the donor individual. Recently, three additional studies used long read PacBio Hi–Fi sequencing of bulk sperm to characterize noncrossover gene conversion tracts (Porsborg et al. 2024; Schweiger et al. 2024; Charmouh et al. 2025). However, at the sequencing depths used by these studies, the long reads covered only relatively small numbers of crossover events that are insufficient for recombination map reconstruction.

Here we present `Hi-reComb`, a new method and software for estimating individual recombination maps from bulk gamete chromosome conformation capture (Hi-C) sequencing data (Lieberman-Aiden et al. 2009; Rao et al. 2014; Oksuz et al. 2021), using a standard Hi-C library preparation protocol. By taking advantage of the long insert sizes delivered by Hi-C, this approach can achieve substantially greater effective coverage by crossover-informative read pairs than has been possible with comparable amount of linked or long read sequencing. Moreover, Hi-C data provide information regarding large-scale structural variation in the donor individual, allowing for simultaneous improvements of the reference genome and better interpretation of the genetic map constructed from this data. First, we show the accuracy of `Hi-reComb` by reconstructing genetic maps from simulated data. Next, to demonstrate the real-world utility of this approach, we constructed and sequenced several sperm Hi-C libraries, used the data to scaffold two reference genomes, and then inferred and evaluated several recombination maps for cichlid and stickleback fish. We show that the maps correspond well to LD-based maps and that the effects of donor haplotype phasing errors are limited. This approach is applicable to any species/individuals that produce at least hundreds of thousands of gametes and are not strongly inbred. Many such species exist; therefore, `Hi-reComb` will be of utility to a broad range of researchers interested in genetics and recombination.

## Materials and methods
### The Hi-reComb approach
Recombination map reconstruction using `Hi-reComb` starts with preparing a standard commercially available Hi-C library. We

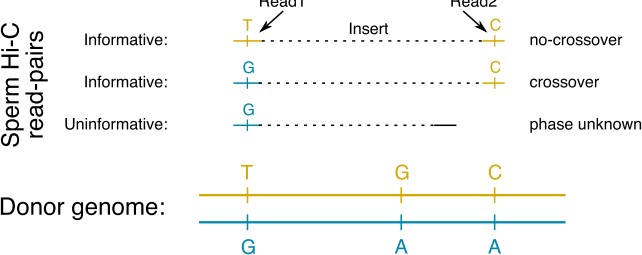

**Fig. 1.** The principle of detecting crossovers with Hi-reComb. The donor genome is shown with heterozygous sites separating the two haplotypes indicated. For each sperm Hi-C read pair, if both fragments match the same donor haplotype, this indicates an absence of crossover (no-crossover). If the fragments match different donor haplotypes, this indicates a crossover between them. Finally, if both fragments do not cover haterozygous sites in the donor genome, the read pair is not informative with regards to recombination.

choose a library based on endonuclease digestion to provide a relatively uniform sequencing coverage (Supplementary Fig. 1). After sequencing and alignment to a reference genome, `Hi-reComb` detects meiotic crossovers by comparison of Hi-C fragments against donor haplotypes (Fig. 1). Because the two reads from each pair of Hi-C fragments originate from the same haploid gamete, cases where the two fragments match different donor haplotypes indicate that a crossover took place between them. We denote this event as $X = C$. Conversely, if the two fragments match the same donor haplotype, this suggests the absence of any crossover between them, denoted $X = N$. The match of Hi-C fragments to either of the two donor haplotypes is determined by sites at which the donor is heterozygous. Therefore, if the Hi-C fragments do not cover at least one heterozygous site each, such a read pair is not informative with regards to crossovers.

`Hi-reComb` estimates crossover likelihoods for each informative read pair i, taking into account the base quality scores output by the sequencer and haplotype phase quality scores output by the phasing algorithm. Let $c_i$ be a parameter indicating if a crossover did ($c_i = 1$) or did not ($c_i = 0$) take place between a pair of reads; thus $\theta = \{c_i | c_i \in \{0, 1\}\}$. We then define a crossover likelihood $l_{ci} = P(X = C | c_i = 1)$ and noncrossover likelihood $l_{ni} = P(X = C | c_i = 0)$, which are calculated as detailed in Supplementary Note 1. Reads with multiple SNPs are used as long as all the SNPs in each read consistently agree on the read pair status ($X = C$ or $X = N$). Then we use the two outermost SNPs for each read pair in the likelihood calculations and all subsequent analyses. While the likelihoods account for base calling and phasing errors, at least as far as they are accurately estimated by the sequencer and the phasing algorithm, other sources of error, e.g. arising in variant calling/filtering or from noncrossover gene conversion events, are unaccounted for. We found that false positive crossover read pairs—i.e. overestimating $l_{ci}$ and underestimating $l_{ni}$—can have substantial impact on the inferred recombination landscapes, creating false "spikes" of recombination. This is true especially for read pairs with short inserts; that is, short genomic distances between the Hi-C fragments.

While false positives are equally likely across the insert length distribution, the proportion of true positives increases linearly with insert length: on average, there are $1,000\times$ more true positives for read pairs with insert size of 1 Mb compared with read pairs with insert size of 1 kb. This is because the further away two loci are from each other along the chromosome (the greater the insert length), the higher the probability that a crossover really occurred between these two loci (i.e. the greater the true positive

rate). To estimate the overall false positive rate and to apply a correction to the likelihoods, we first use likelihoods of long-insert read pairs (default: >1 Mb), where the ratio of true to false positives is highly favorable, to estimate $r_l^*$, the average per-bp crossover rate for each chromosome. Then we calculate a correction factor by evaluating the crossover likelihoods of short-insert read pairs (default: <1 kb) against this baseline. Formally, the correction factor $f$ is defined as $f = \frac{\sum_i l_{ci} - \sum_i d_i * r_l^*}{n_s}$ where $i$ sums over all $n_s$ short-insert informative read pairs with $X = C$ and where $d_i$ is the insert size of the read pair $i$. While the defaults will be suitable in most settings, the thresholds for defining short- and long-insert read pairs can be adjusted using the `–FPrateThresholds` option, which may be useful when analyzing genomes with very small or large chromosomes.

To calculate crossover probabilities, we incorporate the corrected likelihoods together with insert length dependent priors in a Bayesian framework. We assign priors to read pairs under the assumption that the recombination rate is uniform along each chromosome. Therefore, the longer the insert length $d_i$ the higher is the prior for each informative read pair $i$. The prior probability of crossover $p_{ci}$ for each informative read pair $i$ is then simply the product of $r_l^*$ and of the insert length:

$$p_{ci} = r_l^* * d_i$$

Finally, the posterior crossover probability for each informative read pair $i$ where $X = C$ is:

$$\mathbb{P}(c_i) = \frac{(l_{ci} - f) * p_{ci}}{(l_{ci} - f) * p_{ci} + (l_{ni} + f ) * (1 - p_{ci})}$$

For read pairs that do not indicate crossovers, we set $l_{ci} = P(X = N | c_i = 1) = 0$ and $l_{ni} = P(X = N | c_i = 0) = 1$. This is highly "conservative" in terms of avoiding any risk of introducing additional false positives. While it results in false negatives—i.e. underestimating $l_{ci}$ and overestimating $l_{ni}$—this approach does not affect the shape of the estimated recombination landscapes because these false negatives are randomly distributed along the chromosomes.

To reconstruct the recombination map, we initially divide the crossover probability of each Hi-C read pair uniformly along its insert length. That is, we assign the same probability to all basepairs between the informative sites that determine the crossover. Then, we construct an initial recombination map by dividing the sum of crossover and noncrossover probabilities at each bp of the chromosome. While this initial map reflects crossover probabilities of individual read pairs, our goal is to produce a recombination map that integrates probabilities over all read pairs. To achieve this, `Hi-reComb` employs an Expectation-Maximization (EM) procedure (Dempster et al. 1977), which is very similar to the EM procedure employed by (Halldorsson et al. 2019). For a detailed description of the initial map construction and of our EM procedure see Supplementary Note 2.

The core algorithm is supported by several practical heuristics that we found improve its overall performance. The first heuristic concerns an "edge effect" whereby the *effective coverage*—i.e. the number of informative read pairs that span a given genomic interval—drops toward chromosome ends (Supplementary Fig. 1). Reduced effective coverage leads to increased sampling noise, which propagates through the EM algorithm and can affect genomic regions far beyond the chromosome edges. To alleviate this problem, we have introduced a minimum effective coverage

cutoff around chromosome edges, whereby the genomic intervals beyond the cutoff limits are excluded from the EM procedure and from the resulting recombination maps. Second, we eliminate imbalanced SNP and their associated read pairs from consideration before starting the recombination map reconstruction. This reduces errors arising from incorrect variant calling and/or phasing, e.g. due to mismapping of reads. By default, a SNP is considered imbalanced if it bounds two or more read pairs that indicate crossovers ($X = C$), while none of the read pairs it bounds indicate an absence of a crossover ($X = N$). The thresholds can be adjusted using the `--imbalancedSNPs` option. Finally, the third heuristic involves adjusting crossover likelihoods to consider the probability of double crossovers. In this calculation, the probability $p_d$ of two crossovers is given by a Poisson mass function: $p_d = \lambda^2 * e^{-\lambda}/2$, where $\lambda = r_l^*$. Because of crossover interference (Chuang and Smith 2022), this adjustment is limited to for long-insert read pairs (default >1 Mb). The length threshold will be suitable in most settings but can be adjusted depending on the extent of crossover interference in the species of interest (Ernst et al. 2024) by using the `--minDforDoubleCrossovers` option.

## Implementation

The `Hi-reComb` package is efficiently coded in C++, does not have any external dependencies, and is straightforward to install, compile and use. It is open source and freely available from https://github.com/millanek/Hi-reComb. `Hi-reComb` currently contains two core modules and two additional utilities. The first core module, `FindInfoPairs` processes aligned Hi-C read pairs to find the pairs that are informative with regards to crossovers (Fig. 1). It takes as an input a set of phased heterozygous sites in the HapCUT2 format (Edge et al. 2017) and Hi-C read pairs in SAM/BAM format (Danecek et al. 2021) and outputs only pairs of Hi-C reads that cover at least one phased heterozygous site each. The second core module, `RecombMap` performs recombination map inference as described above, with run times of the order of minutes for each chromosome. In addition, `RecombMap` provides a bootstrap option to evaluate uncertainty, whereby informative Hi-C read pairs are resampled with replacement to estimate additional genetic maps. This option also allows taking an average across the bootstraps, which tends to result in smoother maps than a single run estimate.

The `Hi-reComb` package also includes a tool to simulate informative Hi-C read pairs reflecting a known recombination map. This `Simulate` utility enables users to evaluate the accuracy of recombination map inference for a given effective coverage, error rate, insert-size distribution, and map profile. The utility matches the insert-size distribution of the simulated pairs to real Hi-C read pair dataset provided as an input. Each simulated read pair is placed randomly onto a chromosome, and its crossover status (i.e. $X = C$ or $X = N$) is determined by the centimorgan (cM) distance from the input recombination map with an error rate (both false positive and false negative) determined by the `--errorRate` parameter. Read pairs are simulated until reaching a target effective coverage, which is specified by the `-targetCoverage` parameter. The simulated reads are used to reconstruct a recombination map as in the `RecombMap` module. The accuracy of map reconstruction for the given parameters can then be evaluated by comparison with the input map. This procedure can be repeated multiple times using the `-replicates` parameter.

Because the vast majority of read pairs do not have a crossover between them, it is possible to use the sperm Hi-C reads to determine donor haplotype phasing, for example using HapCUT2 (Edge et al. 2017). In this case, no other data is required. This approach

works well in practice, in part because of the robustness of Hi-reComb to errors, including phasing errors, as we demonstrate in the Results section below. However, more accurate phasing can be obtained, for example with Hi-C data from a somatic tissue or by using mother–father–offspring trio data. For the latter purpose Hi-reComb provides the TrioPhase utility, which takes as input a VCF file specifying heterozygous sites in the donor and another VCF file with his parents who have been genotyped at the same loci. The TrioPhase utility outputs phased haplotypes in the same format used by the core modules of Hi-reComb.

## Sperm Hi-C data

We obtained sperm Hi-C data from one individual each of cichlid fish species *Aulonocara stuartgranti* and *Astatotilapia calliptera* (from Lake Malawi), *Neolamprologus multifasciatus* (from Lake Tanganyika), *Astatotilapia nubila* (from the swamps of Lake Victoria), and threespine stickleback *Gasterosteus aculeatus* (from Walby Lake in Alaska), all obtained from laboratory aquarium stocks. A single sexually mature male was sacrificed from each species and his freshly harvested testes were flash-frozen in liquid nitrogen. Upon thawing at room temperature, we cut open the testes and suspended the sperm in ~100 μl of TE buffer. Sperm cells were counted with hemocytometer, aiming for between 100 k and 1 million cells. This was then used as input into the standard Dovetail Omni-C library preparation, following the standard protocol. All libraries were sequenced on the Illumina Novaseq 6000 instrument, obtaining paired end 2 × 150 bp reads.

## Reference genomes and Hi-C scaffolding

We first used to Hi-C reads from *A. calliptera* and from *N. multifasciatus* to produce accurate reference genome assemblies with chromosome-scale scaffolds. For *A. calliptera*, we used as a starting point the fAstCal1.2 genome (GenBank: GCA_900246225.3). This genome assembly was already chromosome-scale, with scaffolding using 10× Genomics Chromium linked reads, BioNano Irys optical maps and two low-resolution genetic maps (Quin et al. 2013; Albertson et al. 2014). However, we found about a hundred disagreements between that assembly and the Hi-C contact map. These discrepancies were corrected manually using the PretextView software (https://github.com/sanger-tol/PretextView). The new manually curated genome was used as a reference for recombination analyses and was deposited as fAstCal1.5 under GenBank accession GCA_900246225.6. For *N. multifasciatus*, we used as a starting point the fNeoMul1.1 genome (GCA_963576455.1), which was not chromosome-scale but was fragmented in 378 large contigs. To produce chromosome-scale scaffolds for this species, we used the YaHS Hi-C scaffolding tool with default parameters (Zhou et al. 2022). The new scaffolded genome fNeoMul1.2 (GCA_963576455.2) was then used for recombination analyses. Finally, for *G. aculeatus* we used the stickleback v5 reference under GCA_016920845.1.

## Alignment and Hi-C contact maps

Hi-C reads were mapped to the reference genomes using bwa mem v 0.7.17 (Li 2013) with the -5SP and -T0 options. To generate Hi-C contact maps, we used the pairtools (Open2C et al. 2024) software (v. 1.1.0), using the --min-mapq 30, --walks-policy 5unique, --max-inter-align-gap 30, and --chroms-path options for the parse command, and using the dedup command with default parameters. The "pairs" files were then used as input into the juicer_tools (v 1.22.01) pre command from the Juicebox package (Durand et al. 2016) which was also used for contact map visualization.

## Variant calling, filtering, phasing

To remove duplicates for all purposes other than the Hi-C contact maps above, we used the MarkDuplicates command from the picard package (v 2.26.6) with the option REMOVE_DUPLICATES=true. Variant calling was done separately for each individual with bcftools v.1.16 (Danecek et al. 2021) using mpileup --count-orphans -Ou output piped into the call program with -mv -Oz options. For variant filtering we used a mappability mask, whereby we broke down the genome into overlapping k-mers of 150 bp (matching the read length), mapped these k-mers back to the genome, and masked all sites where fewer than 90% of k-mers mapped back to their original location perfectly and uniquely, using the SNPable tool (http://lh3lh3.users.sourceforge.net/snpable.shtml). Next, we filtered variants based on sequencing depth, with limits based on examining the coverage histogram for each sample: removing variants with depth ≤ 12 and ≥ 75 for *A. calliptera*, ≤ 35 and ≥ 140 for *A. stuartgranti*, ≤ 20 and ≥ 120 for *N. multifasciatus*, ≤ 35 and ≥ 120 for *A. nubila*, and ≤ 8 and ≥ 100 for *G. aculeatus*. Finally, we applied the following hard filters, removing variants where any of these applied: %QUAL < 20, MQ < 40, MQ0F > 0.4, RPBZ < −5.0, or RPBZ > 5.0. After variant filtering, we kept only biallelic SNPs with heterozygous genotypes. To estimate the allelic phase of these SNPs, we used hapcut2 (v 1.3.4). After dividing the VCF files and the alignment bam files per chromosome, for each chromosome we ran first the extractHAIRS and then the hapcut2 commands, both with the --hic 1 option.

## Stickleback trio-based phasing—data and processing

The *G. aculeatus* sperm donor individual originated from an aquarium F1 cross from wild-caught parents. We sequenced the DNA of his parents to triophase the heterozygous SNPs in this individual using Mendelian inheritance logic (trio phasing). The sequencing was done on the Illumina Novaseq 6000 instrument with 2 × 150 bp reads as a part of a larger DNA sequencing of 24 individuals, each at approximately 25× coverage. For processing this dataset we used alignment with bwa mem v 0.7.17 (Li 2013) with default options, followed by the MarkDuplicates command from the picard package (v 2.26.6) with default options. Then we used GATK v 4.2.3 (DePristo et al. 2011) to call variants, using HaplotypeCaller in GVCF mode for each individual separately followed by joint genotyping using GenotypeGVCFs with the --include-non-variant-sites option. For variant filtering, we generated a callability mask to identify and filter out regions where we were unable to confidently call variants. This included: (i) sites determined by overall read depth cutoffs based on examining a depth histogram (≤ 300 and ≥ 700 on autosomes; ≤ 200 and ≥ 600 on the X chromosome), (ii) sites where >6 individuals had missing genotypes, (iii) sites identified by GATK as low quality (with the LowQual tag) and (iv) sites with poor mappability. The mappability mask was determined in the same way as for the Hi-C analyses above: we broke down the genome into overlapping k-mers of 150 bp (matching the read length), mapped these k-mers back to the genome, and masked all sites where fewer than 90% of k-mers mapped back to their original location perfectly and uniquely. Finally, we used several hard filters based on GATK best practices, specifically focusing on overall genotype quality (QUAL < 20), mapping quality (MQ < 40), mapping strand bias (FS > 40), variant quality normalized by depth (QD < 2) and excess heterozygosity when compared with Hardy–Weinberg equilibrium (ExcessHet > 40).

**Table 1.** Sperm Hi-C data overview.

| Species | Raw coverage | Cis >1 kb reads pairs (millions) | Heterozygous sites per kb | Effective coverage (chr 1) |
|---|---|---|---|---|
| *A. calliptera* | 92× | 18.7 | 1.16 | 292× |
| *A. nubila* | 140× | 48.2 | 1.55 | 3,587× |
| *A. stuartgranti* | 185× | 60.1 | 0.94 | 1,253× |
| *G. aculeatus* | 270× | 2.6 | 2.61 | 655× |
| *N. multifasciatus* | 165× | 40.4 | 1.55 | 1,369× |

## Stickleback trio-based phasing—Hi-reComb trio phasing module

We used the utility `TrioPhase` from Hi-reComb to estimate the haplotype phase of the *G. aculeatus* sperm donor individual using mother-father-offspring trio genotype calls and simple Mendelian logic. For each heterozygous SNP in the offspring, the utility first checks that both alleles are present in the parents at the same site. If both parents are heterozygous then the SNP cannot be phased because the parental genotypes are not informative. In all other cases, the allele inherited from the first parent is added to haplotype 1 and the allele inherited from the second parent is added to haplotype 2. The phased haplotypes are output as a single block in the `hapcut2` format.

## Stickleback LD-based map

We used a dataset of 334 Alaskan stickleback individuals sequenced to mean 19.6× coverage (min: 13.4×; max: 25.8×) on the Illumina Novaseq 6000 instrument with 2×150 bp reads. Alignment, variant calling, and variant filtering was done in the same way as for the trio-based phasing dataset (see above), except that the overall depth filter was set at $\leq 4,000$ and $\geq 8,000$ on autosomes; $\leq 3,000$ and $\geq 7,000$ on the X chromosome and the maximum number of missing genotypes to 66 (i.e. $\leq 20\%$ missingness).

This dataset included 23 individuals from Walby Lake, the same population as the sperm donor. To estimate changes in effective population size ($N_e$) through time, we used smc++ v.1.15.4 (Terhorst et al. 2017), with the commands: vcf2smc → estimate. Then we used the pyrho (Spence and Song 2019) software to infer recombination rates along the genome based on patterns of LD. To build likelihood tables for pairs of biallelic sites, we used the `make_table` command with demographic history as inferred by smc++, and the Moran approximation specified by the `--approx` and `–moran_pop_size N` flags where $N$ equals 1.5× the number of haplotypes. This was followed by the pyrho `optimize` command to infer the recombination maps with a window size of 50 SNPs and block penalty of 15.

## Results
### Sperm Hi-C datasets, contact maps, and genome scaffolding

Key characteristics of the sperm Hi-C datasets presented in this manuscript are summarized in Table 1. A statistic that has a crucial effect on recombination inference is the *effective coverage*. Effective coverage reflects the total length of DNA segments that can be assessed for presence of crossovers and is determined not only by the depth and quality (e.g. the insert-size distribution, the proportion of PCR duplicates) of the Hi-C library but is also substantially influenced by the heterozygosity of the donor individual. The greater the heterozygosity, the greater is the chance that each of the two fragments of a Hi-C read pair covers a heterozygous site and thus is informative, as illustrated in Fig. 1. This

effect is clearly seen in the *G. aculeatus* dataset, where the Hi-C library was of relatively low quality and contained only ~2.6 million read pairs mapping to the same chromosome with >1 kb insert, an order of magnitude lower than all the other samples, likely due to the much lower amount of sperm cells used as a starting material. Despite this, the effective coverage for *G. aculeatus* is comparable to the other datasets.

The use of the Hi-C data for scaffolding of the *N. multifasciatus* genome resulted in 863.6 Mb (98.2%) of sequence being assigned to 22 chromosomes, while the remaining 67 unplaced scaffolds comprise 16 Mb of sequence. The *N. multifasciatus* Hi-C contact map mapped to this new fNeoMul1.2 assembly (GCA_963576455.2) is shown in Supplementary Fig. 2. The use of the Hi-C data for scaffolding of the *A. calliptera* genome resulted in 863.0 Mb (98.3%) of sequence being assigned to 22 chromosomes, while the remaining 122 unplaced scaffolds comprise 14.7 Mb of sequence. The *A. calliptera* Hi-C contact map mapped to this new fAstCal1.5 assembly (GCA_900246225.6) is shown in Supplementary Fig. 3a and examples of how disagreements between the Hi-C contact map and the previous version of the assembly were resolved are shown in Supplementary Fig. 3b. For *A. calliptera*, the chromosome count is as expected, based on known karyotypes of Lake Malawi cichlids (Poletto et al. 2010; Conte et al. 2019), and matches the previous assembly. On the other hand, karyotypes of other species of the cichlid tribe Lamprologini, to which *N. multifasciatus* belongs, showed only 21 chromosomes (Ozouf-Costaz et al. 2017); therefore, our results reveal previously unknown chromosome number polymorphism in this tribe.

Hi-C contact maps provide information regarding large-scale structural variation present in the donor individual. Perhaps the most prominent of these is an inversion in *A. stuartgranti* with respect to the *A. calliptera* reference, located in the middle of chromosome 2 (~12.7 Mb–16.7 Mb), corresponding to the "small" inversion previously reported by (Blumer et al. 2025). Notably, we found that this inversion is surrounded by an extended region of very low recombination in *A. stuartgranti* (Supplementary Fig. 4), illustrating how sperm-based Hi-C contact maps and recombination maps can be used together to better understand the interaction between structural variation and recombination.

### Hi-reComb recombination inference from simulations

To evaluate the accuracy of recombination map inference with Hi-reComb, we used the `Simulate` utility and explored how the performance is influenced by key parameters: the error rate and the effective coverage. For each run, we supplied the Hi-reComb `Simulate` utility with a reference map, simulated 10 replicate Hi-C datasets from this map, and then ran recombination map inference for each replicate.

Figure 2a shows an example of 10 replicate maps reconstructed from simulations with 1% error rate and 3,000× effective coverage, with chr 2 of *A. stuartgranti* as a reference map. With these parameters, the reconstructed maps showed correlation with

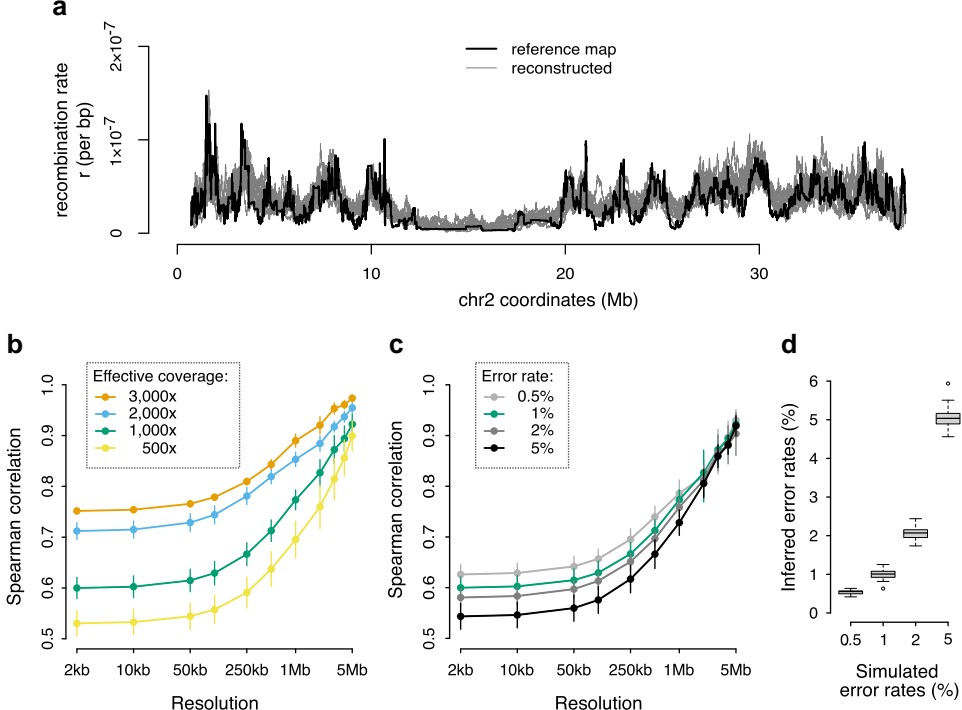

**Fig. 2.** Hi-reComb recombination map inference from simulated data. a) A comparison of a reference map against 10 maps reconstructed by Hi-reComb from simulated data at 3,000× effective coverage and 1% error rate. b) The dependence of accuracy of recombination map reconstruction on the effective coverage. c) The dependence of accuracy of recombination map reconstruction on the error rate. d) The accuracy of the estimates of error rate, i.e. the correction factor *f*, by Hi-reComb.

the truth of between 0.75 and 0.97 (orange line in Fig. 2b), depending on the resolution at which we measured the correlations. In subsequent runs, we varied the effective coverage between 500× and 3,000×, matching approximately the range of coverage found in our empirical datasets (Table 1), while keeping the error rate at 1%. The results, shown in Fig. 2b, revealed that decreasing the effective coverage reduces the accuracy of inferred maps, as expected. However, even at the relatively low effective coverage of 500×, the reconstructed maps show substantial positive correlation with the truth—on average 0.53 at 2 kb scale and 0.90 at 5 Mb scale.

Next, we explored the impact of the error rate on recombination map inference. We fixed the coverage at 1,000× and varied the error rate from 0.5% to 5%. We found that at the high error rate of 5%, Hi-reComb still infers maps with highly positive correlation with the truth (0.54 at 2 kb; 0.92 at 5 Mb), which is only a fraction lower than at the 0.5% error rate where the correlations are 0.63 at 2 kb and 0.93 at 5 Mb scale (Fig. 2c). The resilience to a relatively high degree of error is partly due to the ability of Hi-reComb to accurately infer the error rate from the data: at 1,000× effective coverage, all inference runs reported error rate estimates within a very narrow range of the truth (Fig. 2d).

While the reconstructed recombination landscapes correlate well with the reference map, we note that the recombination rate tends to be overestimated in regions of low recombination as can be seen in Fig. 2a in the extended region in the middle of the chromosome. This overestimation of crossover probabilities has an impact the accuracy of inference of the chromosome-wide average rate: we found that the mean per-bp recombination rate (mean *r*) for the reference map was overestimated by between 8.5% and 14%, with higher effective coverage resulting in more accurate estimates (Supplementary Fig. 5).

## Empirical recombination maps

We used Hi-reComb to infer genetic maps for the five datasets described in Table 1. Error rates estimated from these empirical datasets were within the range where simulations demonstrated reliable performance, with means between 0.8% and 1.9% in cichlids and 0.6% in stickleback (Fig. 3a). The highest error rates of over 4% were found in cichlids on chromosome 3, which contains by far the most highly repetitive sequence where variant calling is difficult (Supplementary Fig. 6). This, along with the lower error rate found in sticklebacks, which have a genome with a much lower proportion of repetitive elements than cichlids (Supplementary Fig. 6), shows the extent to which error rates are affected by miscalled SNPs in repetitive regions of the genome. Among the cichlids, we see a link between error rates and effective coverage, with the low-coverage *A. calliptera* having the highest error rate, followed by the medium coverage *A. stuartgranti* and *N. multifasciatus*, and the lowest error rate among cichlids is in the high-coverage *A. nubila*.

The inferred cichlid recombination maps varied in length from 1,015 centimorgan (cM) for *A. calliptera* to 2,660 cM for *A. stuartgranti*, which falls both below and above two previously published sex averaged pedigree-based maps for Lake Malawi cichlids that had lengths of 1,453 cM (Albertson et al. 2014) and 1,935 cM (Quin et al. 2013). For stickleback, the inferred map had a length of 1,758 cM which is somewhat above the 1,184 cM previously reported for a sex averaged (Roesti et al. 2013) and 1,206 cM for a male specific (Kivioja and Rastas 2024) pedigree-based maps (Fig. 3b). Mean recombination rates can differ substantially between males and females (a phenomenon known as heterochiasmy) (Sardell and Kirkpatrick 2019), the extent of which has recently been quantified for 40 fish taxa, showing that the magnitude and direction of the male vs. female differences vary substantially across species (Kivioja and Rastas 2024).

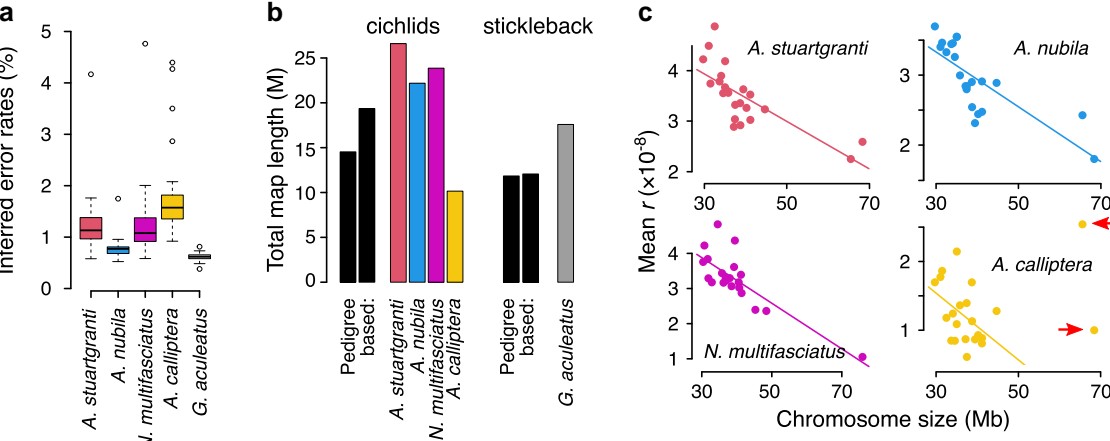

**Fig. 3.** An overview of Hi-C inferred maps. a) The error rates estimated by Hi-reComb from empirical data. b) Recombination map lengths in Morgans (M) compared against three previously published pedigree-based maps (for cichlids, left: (Albertson et al. 2014); right: (Quin et al. 2013); for stickleback, left: (Roesti et al. 2013); right: (Kivioja and Rastas 2024). c) The negative relationship between the chromosome size and the mean recombination rate in cichlids. The two large chromosomes that were excluded in *A. calliptera* as outliers are highlighted with arrows.

We note that the absolute map lengths from Hi-reComb should be interpreted with caution for at least two reasons. First, differing levels of somatic cell "contamination" across samples would affect the overall rate comparisons. While we tried to separate sperm from the surrounding tissues, examination under the microscope showed that our biological samples contained a small number of somatic cells, and we did not conduct further cell purification or sorting. Second, the edge effect, whereby effective coverage decreases toward the ends of each chromosome (see Methods), results in the map ending before the physical chromosome end can have different impact across individuals. Sparse sampling of crossovers near chromosome ends also affects pedigree-based maps (Peñalba and Wolf 2020). The extent of the effect on absolute map lengths depends primarily on the amount of recombination concentrated in subtelomeric regions of chromosomes. Our results for individual chromosomes indicate that such underestimation may be substantial especially for the low-coverage *A. calliptera* individual and for some stickleback chromosomes where the maps have genetic lengths <50 cM (corresponding to one recombination event per chromosome) (Supplementary Fig. 7a). Examination of individual chromosomes also revealed an interesting case where a low recombination rate on N. multifaciatus chr 7 could be explained by incompatibilities between divergent admixture-related haplotypes (Supplementary Fig. 7a and b).

It is known that in many species, there is a negative association between the chromosome length (in Mb) and the mean $r$ (Haenel et al. 2018; Brazier and Glémin 2022). Consistent with these previous studies, our results also indicate a strong negative correlation in both cichlids and in stickleback. In cichlids (Fig. 3c), the negative link was very clear across all chromosomes for *A. stuartgranti* ($r^2 = 0.55$; $p = 7.9 \times 10^{-5}$), *A. nubila* ($r^2 = 0.59$; $p = 3.1 \times 10^{-5}$), and *N. multifasciatus* ($r^2 = 0.64$; $p = 7.1 \times 10^{-6}$). In *A. calliptera*, the two large chromosomes (chr 3 and chr 7) were outliers and the negative association between chromosome length and the mean recombination rate was present only if these two chromosomes were excluded (complete dataset: $r^2 = 0.02$; $p = 0.53$; outliers excluded: $r^2 = 0.20$; $p = 0.049$). In stickleback, the negative association between the chromosome length and mean $r$ was very strong in maps based on the trio phasing, which will be described below ($r^2 = 0.71$; $p = 3.0 \times 10^{-6}$; Supplementary Fig. 8).

To estimate uncertainty in reconstructed recombination landscapes, `Hi-reComb RecombMap` includes a bootstrap procedure whereby informative read pairs are resampled with replacement. As an example, Fig. 4a shows a recombination landscape for chr 4 of *A. stuartgranti* with 95% confidence intervals (95% CIs) estimated based on 50 bootstrap replicate runs. Importantly, the bootstrap estimates facilitate comparisons among recombination landscapes. We define the areas where 95% CIs of two maps do not intersect as areas of significant recombination rate differences, or $\Delta(r)$ regions. To illustrate this functionality, Fig. 4b shows comparisons between the maps of *A. stuartgranti* and *A. nubila*. Overall, we found that $\Delta(r)$ regions between these two species comprised 62.3 Mb of sequence, or 7.23% of the genome, with variation across chromosomes between 3.1 and 14.7%. To account for the fact that mean rates differ between the two maps, we also normalized the means before calculating the $\Delta(r)$ regions. On these mean-normalized maps, $\Delta(r)$ regions comprised 45.7 Mb of sequence, or 5.30% of the genome, with variation across chromosomes between 2.8% and 11.0%.

## Comparisons with trio phasing and with LD-based maps

Our standard workflow uses the same Hi-C dataset for both haplotype phasing and for recombination map inference (see Methods). To compare this approach with independent trio-based phasing, we took advantage of the fact that the *G. aculeatus* donor individual was bred in an aquarium from known parents. We obtained short read whole-genome data from the parents and, after variant calling and filtering, we used `Hi-reComb TrioPhase` to obtain haplotype phase information for ~840 thousand SNPs across the 20 stickleback autosomes. We found that this approach reduced the crossover (false positive/negative) error rate estimated by `Hi-reComb` by more than a third, down to below 0.4% for the *G. aculeatus* data (Fig. 5a). This result suggests that about a third of the errors in the hapcut2-phased datasets arose due to incorrect phasing.

To evaluate the accuracy of the recombination maps inferred by `Hi-reComb` from the hapcut2 and trio-phased datasets, we compared both against a recombination map obtained for the same stickleback population (Walby Lake, Alaska) using LD patterns in an independent population genetic dataset (see Methods). The LD-based map does not represent the ground truth, e.g. due to

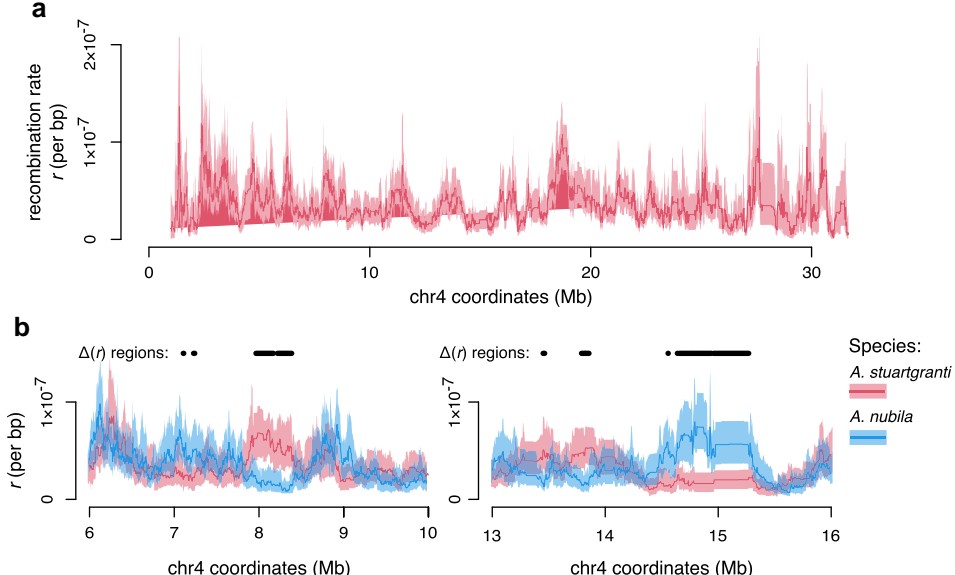

**Fig. 4.** Bootstrap facilitates comparisons between recombination landscapes. a) Chromosome 4 of *A. stuartgranti* with recombination landscapes based on the mean of 50 bootstrap replicates (thick line) and 95% confidence intervals (shaded areas). b) Examples of comparisons between the maps of *A. stuartgranti* and *A. nubila*, highlighting areas of significant recombination rate differences between the maps (the Δ(*r*) regions).

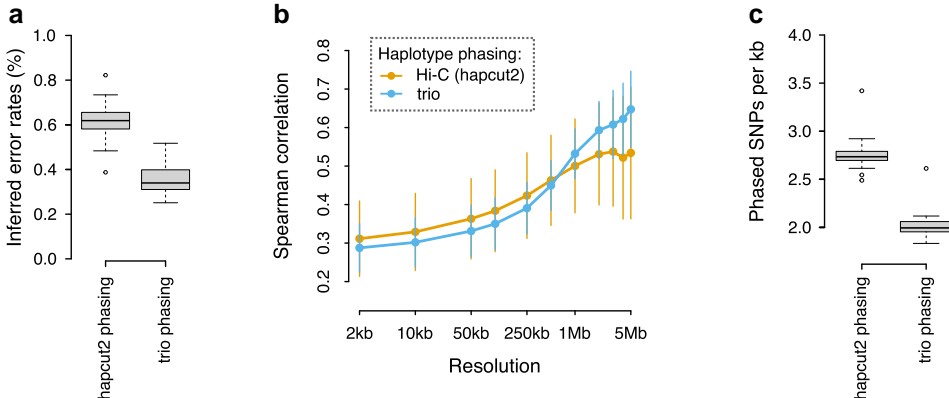

**Fig. 5.** Hapcut2 vs. trio phasing of stickleback data. a) The error rates estimated by Hi-reComb for the hapcut2 and trio-phased *G. aculeatus* datasets. b) Correlations between gamete-based Hi-reComb maps and an LD-based map from the same stickleback population. c) The average density of SNPs phased with the hapcut2 approach and the trio approach. Each datapoint corresponds to one chromosome.

averaging across sexes and through time inherent in LD-based methods. However, we can nevertheless draw tentative conclusions from the map correlations, assuming that incorrect phasing increases noise and this would result in a decreased correlation against the LD-based map. As expected, given its lower error rate, the trio-phased dataset delivers better correlations with the LD-based dataset at resolutions ≥1 Mb (Fig. 5b). However, surprisingly, the pattern changes at finer resolutions (2 to 500 kb), where the hapcut2-phased dataset seems to deliver a more accurate recombination map. The better correlations at fine resolution are likely explained by the fact that hapcut2-based phasing, although less accurate, delivers a greater density of phased heterozygous sites along the genome and thus more fine-scale information (Fig. 5c).

## Discussion

Mapping the distribution of meiotic recombination along chromosomes is a crucial step in many genomic analyses. The Hi-reComb

software provides a new, straightforward, and cost-effective approach for inferring recombination maps, based on sequencing of a Hi-C library from gametes from a single individual. In this manuscript we show results based on fish sperm, but the method will be applicable to a broad range sexually reproducing species. Nevertheless, at least two factors can limit the applicability of Hi-reComb. First, it is necessary to obtain a relatively large number of gametes from a single individual, in the range between 100 thousand and 1 million for the Hi-C protocol we used. While this is easily achievable for males of many larger species (e.g. most vertebrates), the cell count requirement will be challenging to fulfil for males of smaller species and almost always for females who rarely produce such large numbers of gametes. Second, it is necessary that the donor individual has sufficient heterozygosity. We have demonstrated that heterozygosity of ~1 SNP per thousand basepairs is sufficient (Table 1), a value that is toward the lower end of nucleotide diversity range across sexually reproducing species (Leffler et al. 2012; Romiguier et al. 2014). Therefore, individuals

from most natural populations will be sufficiently heterozygous. However, the heterozygosity requirement will pose a limitation for obtaining recombination maps from individuals who are inbred in nature or due to human manipulation.

`Hi-reComb` is well suited for comparisons of recombination among individuals of the same or closely related species, facilitated by the bootstrapping option as illustrated in Fig. 4. It will be interesting to learn more about how much variation there is across individuals, where current knowledge of recombination landscape variation is limited (Johnston et al. 2016; Peñalba and Wolf 2020), with studies mostly focusing on genome-wide crossover counts (Payseur 2024). An important aspect of interindividual variation in recombination is the difference between sexes, which is known to be considerable, at least in some species (Sardell and Kirkpatrick 2019; Kivioja and Rastas 2024). In this context, it should be noted that the `Hi-reComb` approach is limited to diploid sequences and, therefore, we were not able to obtain the X or Y chromosome maps for *G. aculeatus* in which there are large regions of hemizygosity due to degeneration on the Y chromosome (Peichel et al. 2020).

We used a straightforward protocol to illustrate the potential of `Hi-reComb` for routine recombination map inference. At the same time, it is possible to envisage several improvements to the protocol. For example, cell sorting or purification prior to library preparation that ensures that only gametes are used would deliver recombination rate quantification that is more accurate and comparable across individuals. It could also be beneficial to take into account that chromatin in gametes can be distinct from somatic cells, especially in sperm where chromatin is highly condensed by protamines (Okada 2022) or by specific histone proteins in flowering plants (Buttress et al. 2022). Chromatin decompaction treatment of sperm cells could deliver an even more uniform coverage and a higher quality of the Hi-C library (e.g. fewer duplicates, larger insert sizes).

We envisage that `Hi-reComb` will contribute to our understanding of patterns and of the ultimate causes of recombination rate variation by substantially easing the production of gamete-based recombination maps. Given the strengths and weaknesses of this approach, we also see great potential in combining `Hi-reComb` maps with pedigree- and LD-based maps. Measuring recombination in gametes while assessing which haplotypes are transmitted across generations will shed light on the multifaceted interaction between recombination and selection.

## Data availability

The simulated maps and the reconstructed empirical recombination maps are available through DataDryad at https://doi.org/10.5061/dryad.4f4qrfjns. The code used to analyze the recombination maps and generate the results presented in this paper is on GitHub at https://github.com/millanek/Hi-reComb_paper_analyses. All raw sequence data are available on NCBI under the following accessions: BioProject PRJNA1133007 (sperm Hi-C of cichlids), BioProject PRJNA1192732 (stickleback sperm Hi-C and whole-genome sequences of parents for trio-based phasing), BioProject PRJEB49185 (stickleback population genetic data). The Hi-C scaffolded cichlid genomes are available under GCA_900246225.6 for *A. calliptera* (available through ENA at https://www.ebi.ac.uk/ena/browser/view/GCA_900246225.6) and GCA_963576455.2 for *N. multifasciatus*.

## Supplemental material

Supplemental material available at GENETICS online.

## Acknowledgments

We would like to thank Daniel Jeffries, Simon H. Martin, and Aurora Ruiz-Herrera for helpful discussions, Matthew Chotlos and Adrian Indenmaur for fish husbandry, Carolin Sommer-Trembo and Pamela Nicholson for assistance with laboratory work, and the Welcome Sanger Institute sequencing core for DNA sequencing.

## Funding

This work was supported by a Swiss National Science Foundation (SNSF) award to M.M. (grant: 193464), an SNSF award to C.L.P. (grant: TMAG-3_209309/1), and the US National Institutes of Health (NIH) grants (NIAID 1R01AI123659-01A1) to D.B. and (NIGMS 5R35GM142891-04) to J.N.W.

Conflicts of interest. R.E.G. is a co-founder of Dovetail Genomics.

## Author contributions

M.M., R.D., B.S., and R.E.G. conceived and designed the study; M.M. developed the Hi-reComb software and conducted the analyses; M.T., M.M., N.M., and S.S. prepared the Hi-C libraries; C.Z. performed scaffolding of the fNeoMul1.2 genome; M.T. produced the stickleback LD-based maps; J.N.W. bred stickleback for Hi-C sequencing and trio phasing; J.N.W., C.L.P. and D.B. led the fieldwork obtaining stickleback population genetic data; W.S. and O.S. bred cichlid fish for Hi-C sequencing; M.M. wrote the manuscript with comments and input from other co-authors.

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

*Editor: K. Lohse*