## [Peer Review File · Genetics]

Hi-reComb: constructing recombination maps from bulk gamete Hi-C sequencing

Milan Malinsky, Marion Talbi, Chenxi Zhou, Nicholas Maurer, Samuel Sacco, Beth Shapiro, Catherine Peichel, Ole Seehausen, Walter Salzburger, Jesse Weber, Daniel Bolnick, Richard Green, and Richard Durbin

NOTE: The reviews and decision letters are unedited and appear as submitted by the reviewers.

In extremely rare instances and as determined by a Senior Editor or the EIC, portions of a review may be redacted. If a review is signed, the reviewer has agreed to no longer remain anonymous.

The review history appears in chronological order.

Review Timeline:

Submission Date:	2025-03-06
Editorial Decision:	2025-04-14
Resubmission Received:	2025-06-30
Accepted:	2025-07-17

April 14, 2025

GENETICS-2025-307908

Hi-reComb: constructing recombination maps from bulk gamete Hi-C sequencing

Dear Dr. Malinsky:

Two experts in the field have reviewed your manuscript, and I have read it as well. I am pleased to inform you that, with minor revisions, it is potentially suitable for publication in GENETICS. The reviewers have comments and concerns that need to be addressed in a revised manuscript and have made several constructive suggestions for improving this work. You can read their reviews at the end of this email.

It is most important that you address the following in your resubmission:

- 1) discuss the pros and cons of your method compared to similar recent approaches based on PacBio data (reviewer 1).
- 2) include a comparison with a male only recombination map for stickleback and discuss the effect of heterochiasmy when comparing sperm based and pedigree based maps (reviewer 2).
- 3) Provide a bit more detail and justification for the length thresholds and the ad hoc filters (reviewer 2)
- 4) make the genetic maps and the code that was used to produce them available.

Reviewer 1 also suggests a potentially useful check for overcorrection and I think it could be interesting to comment on possible extensions of Hi-reComb to co-estimate crossover and gene conversion.

We look forward to receiving your revised manuscript. Please let the editorial office know approximately how long you expect to need for revisions.

Upon resubmission, please include:

1. A clean version of your manuscript;
2. A marked version of your manuscript in which you highlight significant revisions carried out in response to the major points raised by the editor/reviewers (track changes is acceptable if preferred);
3. A detailed response to the editor's/reviewers' comments and to the concerns listed above. Please reference line numbers in this response to aid the editors.

Additionally, please ensure that your resubmission is formatted for GENETICS.

<https://academic.oup.com/genetics/pages/general-instructions>

Follow this link to submit the revised manuscript: Link Not Available

Sincerely,
Konrad

Konrad Lohse
Senior Editor
GENETICS

Approved by:
Howard Lipshitz
Editor in Chief
GENETICS

Reviewer #1 :

This paper presents an original method (and its associated software) to estimate recombination maps based on bulk gamete Hi-C sequencing. The approach is very elegant, and proves to be very effective: it is considerably cheaper and less labor intensive than classical pedigree-based maps, while achieving a very good level of resolution. As discussed by the authors, their method presents some limitations (it requires a large number of gametes from a single individual, and hence is not appropriate to quantify recombination in females, and more generally, it might not be applicable to small organisms that produce a limited number of gametes). However, it is perfectly suited to quantify recombination in many species where males produce a large amount of sperm (such as vertebrates). One very interesting feature of this method is that it provides the genetic map of single

individuals, and thus allows the investigation of inter-individual variability in recombination rate.

The paper is very clear and very well written. The authors made their software available on GitHub. Thus, this method is extremely promising for all biologists interested in recombination.

I only have minor comments.

#1 In the introduction, the authors review methods that are classically used to reconstruct recombination maps. They could also mention the new approaches based on long-read sperm sequencing that have been recently proposed (Schweiger et al. 2024, Porsborg et al. 2024), and discuss their respective advantages/disadvantages.

Schweiger R, Lee S, Zhou C, Yang T-P, Smith K, Li S, Sanghvi R, Neville M, Mitchell E, Nessa A, Wadge S, Small KS, Campbell PJ, Sudmant PH, Rahbari R and Durbin R 2024, Insights into non-crossover recombination from long-read sperm sequencing. *bioRxiv* <https://www.biorxiv.org/content/10.1101/2024.07.05.602249v1>

Porsborg PS, Charmouh AP, Singh VK, Winge SB, Hvilsom C, Pelizzola M, Laurentino S, Neuhaus N, Hobolth A, Bataillon T and Schierup MH 2024 Insights into gene conversion and crossing-over processes from long-read sequencing of human, chimpanzee and gorilla testes and sperm. *BioRxiv*: <https://www.biorxiv.org/content/10.1101/2024.07.05.601967v1>

#2 As expected, the authors found that their estimates of recombination rates correlate negatively with chromosome size (figure 3C and S6), and positively with LD-based estimates (Fig. 5B). This indicates that their method properly captures variation in recombination rate within genomes. However, they note some discrepancies between the lengths of their genetic maps and those obtained from pedigree studies (lines 359-371). They discuss the fact that their method could be affected by somatic cell contamination, and hence that the absolute map lengths should be interpreted with caution. As I understand, this artefact should lead to underestimate map lengths. One way to assess whether this potential artefact affected their estimates could be to analyze the distribution of chromosome genetic length. Indeed, because of the constraint of having at least one crossover per chromosome per meiosis, the genetic length of the shortest autosomes is almost always above 50 cM (see Fig. 1 in Fernandes et al. 2018). It might thus be interesting to plot the genetic length of chromosomes vs their size to check to what extent some chromosomes deviate from that rule.

NB: the differences between pedigree-based and Hi-reComb estimates do not necessarily reflect errors of the Hi-reComb methods (pedigree-based genetic maps can also be subject to artefacts, and some difference could reflect true variation in recombination rates)

Fernandes JB, Séguéla-Arnaud M, Larchevêque C, Lloyd AH and Mercier R 2018 Unleashing meiotic crossovers in hybrid plants. *Proceedings of the National Academy of Sciences* 115: 2431-2436. <https://doi.org/10.1073/pnas.1713078114>

#3 Data availability (line 471): it would be nice to provide also the genetic maps that were reconstructed for the five fish species.

Reviewer #2 :

I enjoyed reading the manuscript of Malinsky et al. on the construction of recombination landscapes from bulk Hi-C sequencing. The key conceptual insight is that the long inserts from Hi-C sequencing can be leveraged to get large "effective" depth with relatively small "actual" sequencing depth. This enables the detection of crossover across a large number of sperm cells at a relatively low cost. The authors make Hi-Recomb -- a probabilistic framework to estimate recombination landscape from bulk Hi-C sequencing -- available to the community. Simulations demonstrate a strong positive correlation between estimates from Hi-Recomb and ground truth data. Moreover, there is good correlation between recombination maps constructed from bulk sperm sequencing and LD-based maps, while map lengths estimated from pedigree compare fairly well to the estimates of Hi-Recomb. Overall, Hi-recomb proves to be robust and provides a promising and valuable avenue for future research on recombination landscape dynamics, including/particularly in the context of non-model organisms.

My comments are relatively minor, however, I think they may improve the clarity of the manuscript.

1) Heterochiasmy

My main comment regards the fact that heterochiasmy is not discussed in the empirical section on cichlids and sticklebacks recombination maps.

It is known from the literature that:

> Recombination is greater in females than males on all chromosomes [in *Gasterosteus* sticklebacks], and overall map length is

1.64 times longer in females. The locations of crossovers differ strikingly between sexes.
<https://doi.org/10.1534/g3.118.200166>

1.1) LD-based recombination maps are sex-averaged. How meaningful then is it to compare an LD-based map to the map inferred from sperm data (when comparing Trio-phasing and Hapcut2 phasing)? Is it warranted to take the LD map as "ground truth" for the purpose of the analysis? I think this should at least be commented upon.

1.2) Similarly, when comparing pedigree estimates of map length to the sperm-based ones, would it not be better to compare male-specific map lengths? (if possible). For instance, a sex-specific map length for male (1206 cM) and female (2127 cM) three-spined stickleback is provided in <https://doi.org/10.1101/2024.12.23.630081>. Heterochiasmy may explain the discrepancy between estimates from pedigrees and from sperm data (even if it doesn't seem to be the case for sticklebacks here, where the pedigree map length is expected to be longer than the map length estimated from sperm cells).

1.3) On a similar note, I was wondering: are the estimate of Hi-Recomb correlated to the pedigree estimate of recombination rates for each chromosome?
It seems to be an additional useful comparison.

2) Predictive accuracy

2.1) Using simulation data, the values inferred by Hi-Recomb correlate very well to ground truth data -- it has a great relative accuracy. It would also be interesting to know the absolute accuracy (eg: MSE, or the intercept in the regression). For instance, in Figure 2A, the reconstructed landscape correlates well with the reference map, but estimates tend to be higher than the ground truth recombination rate. As mentioned in the text, Hi-Recomb underestimates l_{ci} overestimates l_{ni} . Recombination rates will thus tend to be overestimated, but it would be good to get the quantitative extent.

2.2) Related: the Simulate module include an error rate for both false positives and false negatives. Are FP and FN equally likely in the simulations? Does that make sense? I expect that FP will vastly outnumber FN in real data.

Line by line:

Line 80: "Hi-reComb estimates crossover likelihoods for each informative read pair i considering potential base-calling errors (base quality scores) and phasing errors (phase quality scores)."

I didn't find this sentence very clear. It may be worth it to clarify that the probability of base calling and phasing error are directly taken into account.

Related: does it assume that informative reads have only 1 snp? Are reads with multiple heterozygous sites thrown away? There will be a non-negligible proportion of reads with >1 variant even when heterozygosity ≥ 0.005 .

Line 99: "Then we calculate a correction factor by evaluating the crossover likelihoods of short-insert read pairs (default: $< 1\text{kb}$) against this baseline."

Could it be explained why the 1kb threshold was used? How do the results change when the threshold is changed?

Related: In Figure 2D, I believe the inferred error rate is literally the correction factor f presented in line 101. Is this right? Would it be worth it to mention/clarify it?

Line 136: "Second, if unexpectedly many read pairs are all bounded by the same SNP and/or if unexpectedly many of these read pairs indicate crossovers (i.e., $X = C$), we eliminate these read pairs and SNPs from consideration before starting the recombination map reconstruction."

It may be worth clarifying in the text what is the cause of such events. Mismapping despite the mappability mask? What is the threshold? Can it be adjusted by the users?

Line 139: "Finally, the heuristic involves adjusting crossover likelihoods for long-insert read pairs (default $>1\text{Mb}$) to consider the probability of double crossovers."

Would it be possible to discuss the motivations behind the 1Mb threshold? It seems relatively low. Is it based on empirical CO interference length?

Can the threshold be adjusted by the user? (as the best choice may be species-dependent)

Line 223: "For variant filtering we used a mappability mask, whereby we broke down the genome into overlapping k-mers of 150bp (matching the read length), mapped these k-mers back to the genome, and masked all sites where fewer than 90% of k-mers mapped back to their original location perfectly and uniquely."

I think it is important to explain how it was done: with which software was used or if it is an in-house script, to make it available. More generally, the code to (re)produce the results in the text is not available online.

Line 288: "Effective coverage reflects the total length of DNA segments that can be assessed for presence of crossovers and is determined not only by the depth and quality of the Hi-C library but is also substantially influenced by the heterozygosity of the donor individual."

What does quality mean here? Will the size distribution of inserts matter as well?

Related: Line 62: "This approach is applicable to any species / individuals that produce at least hundreds of thousands of gametes."

Can this sentence be nuanced (based on the info on line 288)?

Associate Editor Comments:

I. 114 "This is highly conservative." To what extent is the correction affected by gene conversion events and so an overcorrection? For tracts <1kb I_{ci} will be overestimated given that gene conversion is not accounted for explicitly? At least in principle, it should be possible to co-infer the gene conversion rate by modelling the fact that with GC, the probability of haplotype switches depends on d_i in a non-linear way (see Setter et al 2022 for an LD based method using this logic). Is this a possible extension of the method?

Minor comments

I. 19-21 It may be useful to distinguish more clearly between direct (gamete and pedigree based) and indirect (LD based) measures of recombination in the intro. The way the various methods are introduced does not really highlight that distinction.

I. 80 Is it correct to assume that reads with more than one informative heterozygous site (which will be common in genetically more diverse organism) are used (as long as all het sites agree on a read's haplotype assignment). It would be good to state explicitly how these are treated (See also reviewer 2).

I. 26 "... but are influenced by selection occurring over that timeframe" Surely not just selection but all popgen processes that affect patterns of LD.

I. 446 space before the ref

Reviewer Response Letter

GENETICS-2025-307908

"Hi-reComb: constructing recombination maps from bulk gamete Hi-C sequencing"

Dear Konrad, dear Reviewers,

We would like to thank you for your constructive and thoughtful comments which have helped us to improve the manuscript and the software. Please find our detailed responses below (line numbers correspond to the 'clean' document):

Associate Editor: Dr. Konrad Lohse

I am pleased to inform you that, with minor revisions, it is potentially suitable for publication in GENETICS.

It is most important that you address the following in your resubmission:

- 1) Discuss the pros and cons of your method compared to similar recent approaches based on PacBio data (reviewer 1).

We added references to three studies using PacBio Hi-Fi bulk sperm sequencing and a brief summary of the key differences. See our response to reviewer 1.

- 2) Include a comparison with a male only recombination map for stickleback and discuss the effect of heterochiasmy when comparing sperm based and pedigree based maps (reviewer 2).

This has been done. See our response to reviewer 2.

- 3) Provide a bit more detail and justification for the length thresholds and the ad hoc filters (reviewer 2)

First, we have provided additional details in the manuscript, including a new comprehensive Supplementary note 1 on likelihood estimation. Second, although we believe the defaults are sensible in most situations, we added options to Hi-reComb to make it possible for users to adjust the thresholds. See our response to reviewer 2.

- 4) Make the genetic maps and the code that was used to produce them available.

Of course. The genetic maps are now on DataDryad (DOI:

<https://doi.org/10.5061/dryad.4f4qrfjns>) and the R code for the analyses presented in this paper is on GitHub (git: https://github.com/millanek/Hi-reComb_paper_analyses).

Reviewer link:

<http://datadryad.org/share/2o8FGCA3s7SxaPjeUcWTuOclSJjsV9qTyCoY9yERqEY>

Reviewer 1 also suggests a potentially useful check for overcorrection and I think it could be interesting to comment on possible extensions of Hi-reComb to co-estimate crossover and gene conversion.

Reviewer 1:

I only have minor comments.

#1 In the introduction, the authors review methods that are classically used to reconstruct recombination maps. They could also mention the new approaches based on long-read sperm sequencing that have been recently proposed (Schweiger et al. 2024, Porsborg et al. 2024), and discuss their respective advantages/disadvantages.

We added these references, and one additional (Charmouh et al., 2025), with a summary of the main advantages and disadvantages of the methods. The long-read PacBio Hi-Fi approach delivers high resolution of individual events, allowing their detailed characterization. All three PacBio studies took advantage of the high resolution to characterize non-crossover gene conversion tracts, e.g. the length distribution, and position with regard to PRDM9 binding sites.

However, these methods are not practical, at least not yet, for reconstructing recombination maps. One would have to do a huge amount of long read sequencing to observe a sufficient number of events. For example, Schweiger et al. sequenced 15 individuals to about 35x coverage each and observed a total of 4,460 crossovers, which is ~13 events per individual, per chromosome. This is by far insufficient for reconstructing individual genetic maps, let alone at a fine scale. In contrast, the Hi-reComb maps are based on at least hundreds and, in most cases, thousands of crossovers per chromosome per individual. The Hi-C approach delivers high effective coverage, at a fraction of sequencing effort, due to the long insert sizes of Hi-C read pairs (lines 58 – 60).

#2 As expected, the authors found that their estimates of recombination rates correlate negatively with chromosome size (figure 3C and S6), and positively with LD-based estimates (Fig. 5B). This indicates that their method properly captures variation in recombination rate within genomes. However, they note some discrepancies between the lengths of their genetic maps and those obtained from pedigree studies (lines 359-371). They discuss the fact that their method could be affected by somatic cell contamination, and hence that the absolute map lengths should be interpreted with caution. As I understand, this artefact should lead to underestimate map lengths. One way to assess whether this potential artefact affected their estimates could be to analyze the distribution of chromosome genetic length. Indeed, because of the constraint of having at least one crossover per chromosome per meiosis, the genetic length of the shortest autosomes is almost always above 50 cM (see Fig. 1 in Fernandes et al. 2018). It might thus be interesting to plot the genetic length of chromosomes vs their size to check to what extent some chromosomes deviate from that rule.

NB: the differences between pedigree-based and Hi-reComb estimates do not necessarily reflect errors of the Hi-reComb methods (pedigree-based genetic maps can also be subject to artefacts, and some difference could reflect true variation in recombination rates)

There are at least two elements that lead to total map length underestimation in Hi-reComb: 1) somatic cell contamination (if sperm cells are not separated / sorted), and perhaps more importantly 2) the edge effect whereby effective coverage decreases towards the ends of each chromosome and we cut off the ends of the maps where insufficient coverage would lead to very noisy map reconstruction. The amount of recombination (in cM) that is removed from the map in this way depends on how concentrated recombination is in subtelomeric regions of the chromosome. We made these points clearer in the revised paragraph (starting on line 398).

As suggested, we looked at the map lengths of individual chromosomes and found that some for stickleback and many for *A. calliptera* (the individual with the lowest coverage) have map lengths below 50cM. We report these findings in the main text and the new **Figure S7A**. However, despite doing various investigations, we did not find anything conclusive confirming exactly which factors are leading to these decreases beyond what is already in the text. Intuitively, perhaps the very low effective coverage in *A. calliptera* is a factor, but we could not confirm that. Perhaps *A. calliptera* simply has more subtelomeric recombination that was cut off. As hinted at by the reviewer, sparse sampling of crossovers near chromosome ends also affects pedigree-based maps, and there can be other artefacts, and we see multiple chromosomes below 50cM both in Fernandes et al. 2018 and in Brazier and Glémin 2022.

Incidentally, reviewing data for individual chromosomes allowed us to identify an outlier chr 7 in *N. multifasciatus*, which has substantially higher heterozygosity (almost 2x higher than most other chromosomes), while having a much lower recombination rate (new **Figure S7B**). This is consistent with the presence of highly diverged haplotypes of admixture origin, which we proposed could lead to lower recombination rates due to incompatibilities.

#3 Data availability (line 471): it would be nice to provide also the genetic maps that were reconstructed for the five fish species.

DONE

Reviewer: 2

My comments are relatively minor, however, I think they may improve the clarity of the manuscript.

1) Heterochiasmy

My main comment regards the fact that heterochiasmy is not discussed in the empirical section on cichlids and sticklebacks recombination maps.

It is known from the literature that:

> Recombination is greater in females than males on all chromosomes [in *Gasterosteus sticklebacks*], and overall map length is 1.64 times longer in females. The locations of crossovers differ strikingly between sexes. <https://doi.org/10.1534/g3.118.200166>

We thank the referee for pointing out this omission from the empirical section of the paper. We now explicitly discuss the differences between male and female recombination rates (line 393).

1.1) LD-based recombination maps are sex-averaged. How meaningful then is it to compare an LD-based map to the map inferred from sperm data (when comparing Trio-phasing and Hapcut2 phasing)? Is it warranted to take the LD map as "ground truth" for the purpose of the analysis? I think this should at least be commented upon.

We agree with the reviewer that the LD-based maps do not represent the ground truth, which we did state in the original text. Now, based on this reviewer comment, we added the reasons why LD-based maps are not the ground truth (e.g., averaging of both sexes and

across many generations, and because of the indirect nature of LD-based inference). We now also explicitly state in the text that the conclusions should be considered as tentative.

Broad, but by no means perfect, correlations between LD-based / pedigree / sperm-based maps have been observed across many previous studies (see e.g. references in the Peñalba and Wolf review). This is reflected in our results (correlation of ~0.3 to ~0.7 depending on genomic scale). However, importantly, our focus is not on the absolute strength of correlation of Hi-reComb maps vs. LD-based maps. Rather, in Figure 5 we are making a technical comparison of two Hi-reComb maps (with hapcut2-phasing vs. trio-phasing). We would argue that as incorrect phasing increases noise, this would result in a decreased correlation against the LD-based map. We strived to explain this in the revised paragraph (starting on line: 452).

1.2) Similarly, when comparing pedigree estimates of map length to the sperm-based ones, would it not be better to compare male-specific map lengths? (if possible). For instance, a sex-specific map length for male (1206 cM) and female (2127 cM) three-spined stickleback is provided in <https://doi.org/10.1101/2024.12.23.630081>. Heterochiasmy may explain the discrepancy between estimates from pedigrees and from sperm data (even if it doesn't seem to be the case for sticklebacks here, where the pedigree map length is expected to be longer than the map length estimated from sperm cells).

We thank the referee for this suggestion. We were not aware of the highly relevant Kivioja and Rastas preprint that came out late last year. We now cite it twice to highlight their great work on estimating male and female recombination rates across many fish species and we use their male-only stickleback map length in the map length comparison in **Figure 3B**.

1.3) On a similar note, I was wondering: are the estimate of Hi-Recomb correlated to the pedigree estimate of recombination rates for each chromosome? It seems to be an additional useful comparison.

The maps of Kivioja and Rastas (2024) unfortunately are not public yet, only the map lengths are. Therefore, we could do this analysis only for the sex averaged maps. As you can see in Figure R1 below, we found a reasonable per-chromosome correlation in the stickleback data using the Roesti et al. (2013), but in the cichlid data there are outlier chromosomes where the Hi-reComb map is relatively much longer/shorter than the pedigree-based map of Quinn et al 2013. There can be a variety of technical reasons (e.g., the Quinn et al. map was built on a very fragmented and incomplete short read genome assembly) and biological reasons (e.g. we are actually looking at different cichlid species) for the cichlid outliers. For stickleback we are looking at the same species, and the genome assembly was essentially complete already in 2013.

Given the limitations of these old pedigree-based maps, and the focus of this manuscript on presenting the Hi-reComb method, rather than empirical investigations, we would prefer not to delve into this in our manuscript.

Figure R1: Chromosome length correlations between pedigree based and Hi-reComb maps.

2) Predictive accuracy

2.1) Using simulation data, the values inferred by Hi-Recomb correlate very well to ground truth data -- it has a great relative accuracy. It would also be interesting to know the absolute accuracy (eg: MSE, or the intercept in the regression). For instance, in Figure 2A, the reconstructed landscape correlates well with the reference map, but estimates tend to be higher than the ground truth recombination rate. As mentioned in the text, Hi-Recomb underestimates I_{ci} overestimates I_{ni} . Recombination rates will thus tend to be overestimated, but it would be good to get the quantitative extent.

The referee is correct: recombination rates tend to be overestimated by Hi-reComb for the reference map used in Figure 2. We quantified this for the reference maps used for the simulations, and the overestimates vary between 8.5% and 14%, depending primarily on the effective coverage, with higher coverage resulting in more accurate estimates. We have added a paragraph detailing this result to the manuscript: a new paragraph starting on line 367 and a new supplementary **Figure S5**.

2.2) Related: the Simulate module include an error rate for both false positives and false negatives. Are FP and FN equally likely in the simulations? Does that make sense? I expect that FP will vastly outnumber FN in real data.

We believe it makes sense to have a single error rate. We do not see a clear reason to believe that the probability that a read pair is misclassified depends on whether it, in fact, contains or does not contain a crossover. The sources of error are chiefly in base-calling, variant calling, and haplotype phasing.

It is true that FP vastly outnumber FN in real data. This is because the vast majority of read pairs are true negatives, they do not contain a crossover. Therefore, one error rate, i.e. the probability that a read pair is misclassified, leads to different numbers of FPs and FNs. There is no contradiction.

Line by line:

Line 80: "Hi-reComb estimates crossover likelihoods for each informative read pair i considering potential base-calling errors (base quality scores) and phasing errors (phase quality scores)." I didn't find this sentence very clear. It may be worth it to clarify that the probability of base calling and phasing error are directly taken into account.

We see that the sentence was and the entire paragraph were too vague. We revised the paragraph, clarifying that what goes into the calculation of the likelihoods are: a) sequencing quality scores output by the (Illumina) sequencer; and b) haplotype phase quality scores output by the phasing software, which in our case was the hapcut2 software. We also added a new **Supplementary Note 1** with full details of crossover likelihood calculations.

Related: does it assume that informative reads have only 1 snp? Are reads with multiple heterozygous sites thrown away? There will be a non-negligible proportion of reads with >1 variant even when heterozygosity ≥ 0.005 .

Yes, it is important that we clarify this (new lines 91 to 95). Reads with multiple SNPs are used as long as the SNPs all have the same haplotype phase – i.e. if they consistently agree on the read pair status ($X = C$ or $X = N$). Then we use the two outermost SNPs for each read pair in the likelihood calculations and all subsequent analyses.

In the future, it could be explored how to take the presence of multiple SNPs in a read into account in the likelihood calculations, but it is not entirely obvious because the SNPs are not independent, e.g. a mapping error for the read would affect both SNPs in that read.

Line 99: "Then we calculate a correction factor by evaluating the crossover likelihoods of short-insert read pairs (default: < 1kb) against this baseline."

Could it be explained why the 1kb threshold was used? How do the results change when the threshold is changed?

The comparison is <1000bp versus >1Mb. This is sensible because:

- 1) In a typical Hi-C experiment, there are a large number of reads in both categories
- 2) The true positive rate is more than three magnitudes higher for >1Mb read pairs

Due to these reasons, we can estimate the correction factor accurately with these cutoffs, as demonstrated by the analyses of simulated data.

Because of the accuracy of our estimates of f , we do not consider it necessary to systematically explore the effects of changing these thresholds. We tried e.g. <2kb vs. >2Mb and, as expected, the results do not change dramatically.

Nevertheless, we added an option for users to adjust the thresholds. We see that especially adjusting the higher threshold may be useful for species with much smaller or larger than chromosomes than we analysed (the fish chromosome sizes are in the region of 30-40 Mb). We added a note to that effect to the manuscript (line 114).

Related: In Figure 2D, I believe the inferred error rate is literally the correction factor f presented in line 101. Is this right? Would it be worth it to mention/clarify it?

Yes, this is correct. We now state this explicitly in the legend of **Figure 2D**.

Line 136: "Second, if unexpectedly many read pairs are all bounded by the same SNP and/or if unexpectedly many of these read pairs indicate crossovers (i.e., $X = C$), we eliminate these

read pairs and SNPs from consideration before starting the recombination map reconstruction."

It may be worth clarifying in the text what is the cause of such events. Mismapping despite the mappability mask?

What is the threshold? Can it be adjusted by the users?

We clarified the default thresholds for the SNPs to be removed (zero non-crossover read pairs, i.e. $X = N$, and at least two crossover read pairs with $X = C$). The thresholds can now be adjusted by users using the `--imbalancedSNPs` option (stated on lines 151-154).

Yes, the type of false positives eliminated by this heuristic is most likely due to mismapping, although other reasons for variant calling or phasing errors are conceivable (e.g. structural variants in the region). As we do not want to speculate in the manuscript, we do not go into further detail about the sources of these errors.

Line 139: "Finally, the heuristic involves adjusting crossover likelihoods for long-insert read pairs (default >1Mb) to consider the probability of double crossovers."

Would it be possible to discuss the motivations behind the 1Mb threshold? It seems relatively low. Is it based on empirical CO interference length? Can the threshold be adjusted by the user? (as the best choice may be species-dependent)

Yes, the minimum distance threshold is motivated by crossover interference, and the extent can now be adjusted by the users. We have clarified the motivation and the flexibility in the main text (lines 157-161).

The 1Mb default is modest, informed by Fig. 1C in Ernst et al. (2024), which suggests that double crossovers at ~1Mb distance do occur at a non-negligible frequency.

Line 223: "For variant filtering we used a mappability mask, whereby we broke down the genome into overlapping k-mers of 150bp (matching the read length), mapped these k-mers back to the genome, and masked all sites where fewer than 90% of k-mers mapped back to their original location perfectly and uniquely."

I think it is important to explain how it was done: with which software was used or if it is an in-house script, to make it available.

We used Heng Li's SNPable tool (<http://lh3lh3.users.sourceforge.net/snpable.shtml>), which we clarify in the text (line 245-246).

More generally, the code to (re)produce the results in the text is not available online.

The code used to analyze the recombination maps and generate the results presented in this paper is now on GitHub at https://github.com/millanek/Hi-reComb_paper_analyses. We added this link to the Data Availability statement.

Line 288: "Effective coverage reflects the total length of DNA segments that can be assessed for presence of crossovers and is determined not only by the depth and quality of the Hi-C library but is also substantially influenced by the heterozygosity of the donor individual."

What does quality mean here? Will the size distribution of inserts matter as well?

Yes, the insert size distribution is an important factor. Also e.g., proportion of PCR duplicates. We have clarified this in the text (line 310).

Related: Line 62: "This approach is applicable to any species / individuals that produce at least hundreds of thousands of gametes." Can this sentence be nuanced (based on the info on line 288)?

Yes, we added a note that the individual / species should not be strongly inbred (line 70).

Associate Editor – detailed comments:

I. 114 "This is highly conservative." To what extent is the correction affected by gene conversion events and so an overcorrection? For tracts $<1\text{kb}$ I_{ci} will be overestimated given that gene conversion is not accounted for explicitly? At least in principle, it should be possible to co-infer the gene conversion rate by modelling the fact that with GC, the probability of haplotype switches depends on d_i in a non-linear way (see Setter et al 2022 for an LD based method using this logic). Is this a possible extension of the method?

We agree that it would be desirable to account for gene-conversion explicitly, and we understand the reasoning of Setter et al. 2022, and others who develop methods in this direction. However, given the short read lengths of (Illumina) Hi-C sequencing, it is not clear to us how this could be done within the context of Hi-reComb.

Consider the typical scenario for what we call a 'crossover read pair' with one heterozygous site covered by each of the two reads. The 'crossover' classification is inferred by Hi-reComb from the observation that the two reads are out of phase. However, this out-of-phase status could be due to gene conversion, with one of the reads falling within a gene conversion tract. Importantly, this can be the case even if the distance - the insert size - between the two reads is large, e.g. several Mb. As long as one read, or more specifically the heterozygous site it covers, is within a gene conversion tract while the other read is not, the read pair will be (mis-)classified by Hi-reComb as a 'crossover read pair'. And, given this data, we do not see a way to directly distinguish between a true crossover vs. non-crossover gene conversion in this scenario. Our solution, although not ideal, is to account for this possibility implicitly in the 'error rate' when integrating across read-pairs.

More generally, depending on the lengths of gene conversion tracts, read-lengths, and heterozygosity, there will be ways to directly distinguish at least some gene conversion events in some scenarios. But an extension of the Hi-reComb method in this direction would not be trivial.

Minor comments

I. 19-21 It may be useful to distinguish more clearly between direct (gamete and pedigree based) and indirect (LD based) measures of recombination in the intro. The way the the various methods are introduced does not really highlight that distinction.

We edited this paragraph to make that distinction clearer (to the paragraph starting on line 15).

I. 80 Is it correct to assume that reads with more than one informative heterozygous site (which will be common in genetically more diverse organism) are used (as long as all het sites agree on a read's haplotype assignment). It would be good to state explicitly how these are treated (See also reviewer 2).

Yes, they are used, and we now state this explicitly. Please see our response to Reviewer 2.

I. 26 "... but are influenced by selection occurring over that timeframe" Surely not just selection but all popgen processes that affect patterns of LD.

Yes, of course. We added a note to that effect (line 27)

I. 446 space before the ref

DONE

July 17, 2025

RE: GENETICS-2025-308324

Dr. Milan Malinsky
Universitat Bern
Institute of Ecology and Evolution
Baltzerstrasse 12
Bern 3012
Switzerland

Dear Dr. Malinsky:

Congratulations, your manuscript titled "Hi-reComb: constructing recombination maps from bulk gamete Hi-C sequencing" is accepted for publication in GENETICS! Many thanks for submitting your research to the journal.

The reviewers had a few suggestions for improving the manuscript that you may want to consider. You can view their comments at the bottom of this email. In particular, it would be good to clarify/revise the description of figure S4 as suggested by reviewer 2.

To Proceed to Publication:

1. Format your article according to GENETICS style: <https://academic.oup.com/genetics/pages/general-instructions>

2. Ensure that you comply with data and community resource citation guidelines:
<https://academic.oup.com/genetics/pages/general-instructions#Data-Policy>

3. Upload your final files at <https://genetics.msubmit.net>

4. Add oupsupport@scipris.com and genetics.oup@novatechset.com (or the domains @scipris.com and @novatechset.com) to your email program's "safe senders" list. You will be contacted by both at various points during the production process.

Notes:

- Your currently-accepted manuscript (unedited, as submitted, reviewed, and accepted) will be published at GENETICS and deposited into PubMed as an Advance Access article. Notify sourcefiles@thegsajournals.org before signing your license if you do not wish to publish your article via Advance Access.

- We invite you to submit an original color figure related to your paper for consideration as cover art. Please email your submission to the editorial office or upload it with your final files. You can submit a small-sized image for evaluation, and if selected, the final image must be a TIFF file 2513px wide by 3263px high (8.375 by 10.875 inches; resolution of 600ppi). Please avoid graphs and small type.

- After files are sent to Oxford University Press we use SciPris to manage article licensing and payment. If you do not have a SciPris account, you will receive an email from no-reply@scipris.com to sign up to use Oxford University Press' author portal. After logging in, follow the online instructions to sign your license and arrange any payment due.

If you have any questions or encounter any problems while uploading your accepted manuscript files, please email the editorial office at sourcefiles@thegsajournals.org.

Sincerely,

Konrad Lohse
Senior Editor
GENETICS

Approved by:
Howard Lipshitz
Editor in Chief
GENETICS

Review comments (if applicable):

Reviewer #1 :

The authors properly addressed the points that I had raised in my previous review.

I just noticed 2 typos:

line 47: 'demonstrating' -> 'demonstrating'

line 413: 'could be explain' -> 'could be explained'

Reviewer #2 :

Thank you for your responses; I found them compelling. It's great that the methods were clarified, and the addition of chromosome map length makes the method more convincing.

I have one final comment. The inversion shown in Fig. S4 is not heterozygous but homozygous for an inversion relative to the reference assembly. The Hi-C map would show contact at the breakpoint in the central diagonal if it were heterozygous. It's mentioned that it's an inversion relative to the reference, but not that it's homozygous. This might seem like a semantic issue, but initially, I understood the figure as showing the effect of recombination suppression in an inversion heterozygote, which is not the case here. I understand it's just a proof of concept for the method and that recombination suppression isn't the main focus, but tweaking the phrasing could help avoid confusion.